# A ClearSee-Based Clearing Protocol for 3D Visualization of *Arabidopsis thaliana* Embryos

**DOI:** 10.3390/plants10020190

**Published:** 2021-01-20

**Authors:** Ayame Imoto, Mizuki Yamada, Takumi Sakamoto, Airi Okuyama, Takashi Ishida, Shinichiro Sawa, Mitsuhiro Aida

**Affiliations:** 1Graduate School of Biological Sciences, Nara Institute of Science and Technology (NAIST), 8916-5 Takayama, Nara 630-0192, Japan; iabs.hr6@gmail.com; 2International Research Organization for Advanced Science and Technology (IROAST), Kumamoto University, 2-39-1 Kurokami, Chuo-ku, Kumamoto 860-8555, Japan; myamada@kumamoto-u.ac.jp (M.Y.); ishida-takashi@kumamoto-u.ac.jp (T.I.); 3Faculty of Science, Kumamoto University, 2-39-1 Kurokami, Chuo-ku, Kumamoto 860-8555, Japan; t.s613@icloud.com (T.S.); flute.airi0709@gmail.com (A.O.); 4Graduate School of Science and Technology, Kumamoto University, 2-39-1 Kurokami, Chuo-ku, Kumamoto 860-8555, Japan; sawa@kumamoto-u.ac.jp

**Keywords:** clearing, 3D imaging, *Arabidopsis thaliana*, embryo, confocal microscopy, cell wall staining, fluorescent reporter, GFP

## Abstract

Tissue clearing methods combined with confocal microscopy have been widely used for studying developmental biology. In plants, ClearSee is a reliable clearing method that is applicable to a wide range of tissues and is suitable for gene expression analysis using fluorescent reporters, but its application to the *Arabidopsis thaliana* embryo, a model system to study morphogenesis and pattern formation, has not been described in the original literature. Here, we describe a ClearSee-based clearing protocol which is suitable for obtaining 3D images of *Arabidopsis thaliana* embryos. The method consists of embryo dissection, fixation, washing, clearing, and cell wall staining and enables high-quality 3D imaging of embryo morphology and expression of fluorescent reporters with the cellular resolution. Our protocol provides a reliable method that is applicable to the analysis of morphogenesis and gene expression patterns in *Arabidopsis thaliana* embryos.

## 1. Introduction

In plant development, oriented cell division and expansion play essential roles in morphogenesis and pattern formation [1]. The embryogenesis of *Arabidopsis thaliana*, in which a relatively small number of tissues and organs are arranged in a simple pattern, provides an excellent system to study morphogenesis and pattern formation, and many regulatory factors that affect these processes have been identified and studied extensively [2,3]. Moreover, because patterns of cell division and elongation are significantly regular during *Arabidopsis* embryogenesis [4,5,6], their possible roles in development and the underlying mechanisms for oriented cell division and elongation have been an important subject [7,8,9,10].

Because morphogenesis and pattern formation occur not only on the surfaces of the embryo but also in its internal structures (e.g., vascular and ground tissues), a reliable method for visualizing morphological and patterning events that occurs deep inside the embryo is necessary. Tissue clearing is a powerful technique to meet such requirements, and several protocols for clearing plant structures have been reported [11,12,13,14]. Among them, TOMEI-II [13] and ClearSee [11] have an advantage in visualizing gene expression patterns, as these methods well preserve the fluorescence of various fluorescent proteins. Although both methods can be applicable to a wide range of tissue types and to various plant species, whether they can also work with embryos has not been reported. Moreover, a recent application of ClearSee to *Arabidopsis thaliana* ovules [15] confirms that the method has potential to visualize plant internal structures with high quality. Here, we established a protocol to apply the ClearSee method to the embryo of *Arabidopsis thaliana* and to demonstrate that the protocol can visualize cellular arrangement and the signal of various fluorescent reporters in 3D.

## 2. Results and Discussion

### 2.1. Dissection of Embryos

We first applied the original ClearSee protocol [11] to whole seeds with the expectation of visualizing embryos without dissection. However, seeds processed with this protocol exhibited brown color in the endothelium (Figure 1A), preventing us from imaging internal embryos. We therefore decided to manually dissect embryos before applying the protocol.

For dissecting embryos, seeds were first removed from the fruit according to the method described previously [16] except with 7% glucose solution instead of N5T medium. Briefly, each of the valves was slit open using a needle and was partly removed from the fruit by using forceps to expose the seeds. The half-opened fruit was completely immersed in 3 mL of 7% glucose solution in a 35-mm dish, and seeds were excised by using a pair of forceps under a stereo microscope. To avoid floating of the seeds, which made the subsequent embryo isolation difficult, the whole fruit and hence the seeds within were kept submerged during dissection by holding its pedicel with another pair of forceps.

The excised seeds were then subjected to manual dissection of embryos. Within a seed, the embryo is located on the micropyle/chalaza side (Figure 1B). To isolate embryos, the other side of the seed was excised by using forceps and the seed surface around the micropyle was gently pushed several times with the tips of the forceps until the embryo popped out from the open end (Figure 1C). Isolating 5–10 embryos each time, they were collected using a P20 micropipette, which was adjusted to 1–2 μL, and were assembled within a small area in the same 35-mm dish, where the debris produced by dissection was not present. This process was important to avoid losing isolated embryos by mixing them with the debris. Occasionally, small debris may adhere to an embryo, reducing the quality of imaging. Such debris can often be removed by gently scratching it with an eyelash attached to a toothpick.

### 2.2. Fixation and Washing

After tens of embryos were isolated and assembled, they were subjected to fixation and washing by transferring embryos from one solution to another by a P20 micropipette. To minimize the risk of losing embryos during this process, we put a drop of solution (100–200 μL) at the center of a dish instead of filling up the dish with a larger amount of solution. Moreover, embryos often adhered to the bottom of the dish or to the inner surface of the pipette tips when they were in the fixative or the washing buffer, further increasing the risk of losing them. To avoid this, we added a very small amount of ClearSee to each solution (e.g., 0.5 μL per 1 mL) prior to use because ClearSee contained the detergent sodium deoxycholate, which prevented embryos from adhering. When transferring embryos, the volume of the micropipette should be adjusted to 1–2 μL to minimize carryover of the solution.

The isolated embryos were collected from 7% glucose by using P20 micropipette under the stereo microscope and were transferred to a drop of the fixative. The embryos were incubated in the fixative for 10 min at room temperature. We tested 5, 10, 30, and 60 min as the fixation time, which gave essentially the same results in terms of embryo morphology, cell wall staining patterns, and fluorescence of reporters. The fixed embryos were then washed twice by sequentially transferring them to the first and second drops of the washing buffer that were placed in separate 35-mm dishes and by incubating them for 1 min each. Vacuum infiltration, which was described in the original ClearSee protocol [11], was not necessary.

### 2.3. Clearing and Staining

Clearing was carried out by transferring the fixed and washed embryos to 3 mL of ClearSee solution in a 35-mm dish. When the embryos were released from the P20 micropipette, they initially floated at the surface of the solution and then gradually sank until they reached the bottom as infiltration proceeded. The dish was then sealed with parafilm and was kept dark for 1–7 days at room temperature. Embryos at the late stages required longer incubation times compared to those at early stages.

From this step on, the embryos became difficult to see with the progression of clearing. To visualize the embryos for subsequent processing, the illumination settings of the stereo microscope were critical. Off-axis (oblique) illumination [18] gave significantly higher contrast than bright- or dark-field illumination, facilitating monitoring and collection of the embryo samples (Figure 1D–F).

The cleared embryos were then transferred to the staining solution containing Calcofluor White and were kept for 1 h at room temperature. Again, a 100–200 μL drop of the staining solution was used to avoid loss of embryo samples. After staining, the embryos were transferred to ClearSee and kept for 1 h to remove excess of the dye.

### 2.4. Confocal Microscopy

For mounting embryo samples, two pieces of double-sided tape were pasted with an appropriate interval onto a glass slide as spacers. The cleared embryos were mounted in an area between the spacers and covered with a coverslip. Marking the positions of the mounted embryos with a felt-tip pen on the coverslip helped to locate embryos under a confocal microscope.

Figure 2 shows a set of images obtained from an embryo carrying the *DR5rev::GFP* reporter [19]. Z-stack images of 157 serial optical sections with the 0.3-μm interval were acquired (Appendix A) and used for observing a single focal plane (Figure 2A) or for reconstructing 3D image (Figure 2B,C). Both cell walls labelled by Calcofluor White and auxin response marked by the accumulation of endoplasmic reticulum (ER)-localized Green Fluorescent Protein (GFP) are clearly visible, and both patterns of cellular configuration and distribution of the *DR5rev* activity are confirmatory with previous observations [8,20], showing that the cell wall structure and GFP fluorescence are well preserved after the processing with our protocol. Moreover, 3D reconstruction allows for identifying geometrical features of cell morphology and gene expression patterns. The image would also be suitable for quantitative analyses using imaging software such as ImageJ [21] or MorphographX [22].

We next applied our protocol to other reporter lines that express different fluorescent proteins with different cellular localizations from those of *DR5rev::GFP*. The embryos of *pARF5-n3GFP*, which produce a three-tandem repeat of GFP (3xGFP) with the SV40 nuclear localization signal (NLS) [23], give clear signals in regions including provascular tissues at stages from the heart to bending-cotyledon stages (Figure 3A–C), showing that our protocol is also applicable to a nuclear-localized version of the fluorescent protein accumulated in inner tissues. We also tested samples simultaneously expressing GFP and Red Fluorescent Protein (RFP). When we examined embryos producing WUSCHEL (WUS)-3xGFP under the native regulatory sequences of the *WUS* gene (*gWUS-GFP3* [24]) and HISTONE 2B (H2B) fused to mScarlet (an RFP derivative [25]) under the control of *CLAVATA3* (*CLV3*) regulatory sequences [26] (*pCLV3:H2B-mScarlet*), green signals in the L3 layer as well as red signals in the L1–L3 layers of the shoot apex was clearly detectable (Figure 3D). These results show that our modified ClearSee protocol enables visualization of different fluorescent proteins (GFP and RFP) with different tags (ER retention, SV40-NLS, WUS, and H2B) at a range of embryonic stages (heat to bending-cotyledon stages).

## 3. Materials and Equipment

### 3.1. Plant Materials

The *Arabidopsis thaliana DRrev::GFP* and *pARF5-n3GFP* reporter lines were obtained from the Arabidopsis Biological Resource Center (ABRC stock number CS9361 and CS67076, respectively). To construct *pCLV3::H2B-mScarlet*, the *Arabidopsis thaliana H2B* (AT5G22880) coding region was PCR amplified from the wild-type Col-0 genomic DNA using the primer set H2B_F and H2B_R. An additional round of PCR was performed using the attB1 and attB2 primers. The PCR product was then cloned into pDONR221 vector using BP clonase II (Thermo Fischer Scientific, Waltham, USA). The entry clone harboring H2B was PCR amplified using the primer set H2B_entry_F and H2B_entry_R, and the mScarlet coding sequence was PCR amplified from the pmScarlet_C1 [25] (Addgene #85042, Watertown, USA) using the primer set mScarlet_F and mScarlet_R. These two PCR products were integrated using In-Fusion HD Cloning Kit (TaKaRa, Kusatsu, Japan) to give *H2B-mScarlet pDONR221*. In parallel, the SalI-SacI fragment of pBU14 containing 5′ and 3′ regulatory sequences of the *CLV3* gene [26] was transferred to the corresponding sites of pBIN41, a pBIN19-derived binary vector carrying a hygromycin resistance gene, yielding *CLV3p-CLV3t pBIN41*. The *H2B-mScarlet* sequence was amplified from *H2B-mScarlet pDONR221* using the primer set H2B_mScarlet_0343_F and H2B_mScarlet_0343_R and was cloned into the BamHI site of *CLV3p-CLV3t pBIN41* by using NEBuilder HiFi DNA Assembly Master Mix (New England Biolab, Ipswich, USA) to yield *pCLV3::H2B-mScarlet*, which was transformed to plants carrying both *gWUS-GFP3* [24] and *RPS5Ap:5mCUC1-GR* [27]. The sequences of the primers are listed in Appendix A. Plants were grown as described previously [27]. Seeds were surface-sterilized using 10% commercial bleach (Kao Corporation, Tokyo, Japan) and were sown on plates containing half-strength Murashige-Skoog salts, 1% sucrose, and 0.5% gellan gum (Fujifilm Wako Pure Chemical Cooperation, Osaka, Japan). After incubation for 4–7 days at 4 °C in the dark, the plates were incubated in a growth chamber at 23 °C under constant white light. After incubation for 2 weeks, the plants were transplanted onto soil and grown under constant white light or a cycle of 8 h dark/16 h light.

### 3.2. Solutions

The solutions, 7% glucose in water (*w*/*v*), fixative (4% paraformaldehyde in phosphate buffered saline (PBS, *w*/*v*)), washing buffer (PBS; 130 mM NaCl, 7 mM Na_2_HPO_4_, and 3 mM NaH_2_PO_4_; pH 7.0), ClearSee (10% xylitol (*w*/*v*), 15% sodium deoxycholate (*w*/*v*), and 25% urea (*w*/*v*) in water), and staining solution (100 µg/mL Calcofluor White M2R (F3543, Sigma-Aldrich, St. Louis, MO, USA) in ClearSee) were prepared as described previously [11]. For preparing ClearSee, the reagents were first dissolved in the 0.6 volume of water and then the final volume was adjusted by adding extra water.

### 3.3. Equipment

Equipment consisted of stereo microscopes equipped with a transmitted light unit capable of oblique illumination (Stemi 2000-C and Stemi 508 Stand KLAB, Carl Zeiss, Oberkochen, Germany; SMZ1270, Nikon, Tokyo, Japan), the confocal microscope (Leica TCS-SPE, Leica microsystems GmbH, Wetzlar, Germany; Olympus FV3000, Tokyo, Japan), forceps (Dumont #5, Manufactures D’Outils Dumont SA, Montignez, Switzerland), and dishes (IWAKI non-treated 35-mm culture dishes 1000-035, IWAKI, Shizuoka, Japan). For Z-stack image acquisition of confocal microscopy, a 63× oil-immersion and a 40× dry objective lenses were used. Calcofluor White and GFP were excited by 405 nm and 488 nm laser lines, respectively, and were detected using 410–480 nm and 490–540 nm filter settings, respectively, with the sequential line scan mode. Images were processed using Leica LAS-X software (Leica microsystems GmbH, Wetzlar, Germany) and ImageJ [21].

## Figures and Tables

**Figure 1 plants-10-00190-f001:**
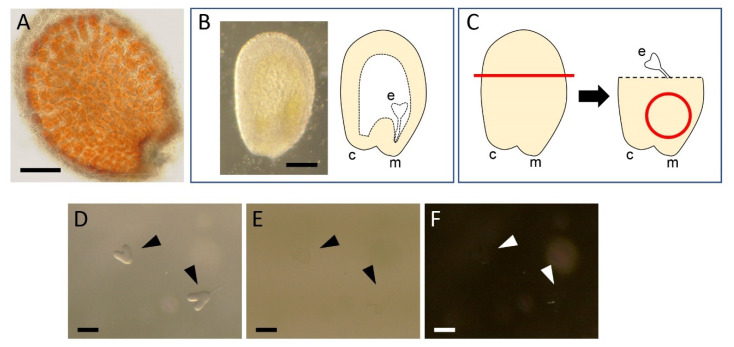
Dissection of *Arabidopsis thaliana* embryos: (**A**) a seed processed with the original ClearSee protocol [11], not transparent and exhibiting a brown color; (**B**) seeds excised from a fruit in 7% glucose solution (**left**) and schematic diagram of its internal structure (**right**); (**C**) the procedure of embryo isolation, where half of the seed is excised along the red line (**left**) and the region around the micropyle marked with a red circle (**right**) is pushed several times until the embryo pops out; and (**D**–**F**) the effects of illumination of a stereo microscope. In ClearSee solution, embryos are clearly visible with oblique transmitted illumination (**D**) whereas they are almost invisible with bright-field (**E**) or dark-field (**F**) illumination. The arrowheads indicate the positions of embryos. c, chalaza; e, embryo; and m, micropyle. Bars = 100 μm. Diagrams in B and **C** are modified from Hughes, 2009 [17].

**Figure 2 plants-10-00190-f002:**
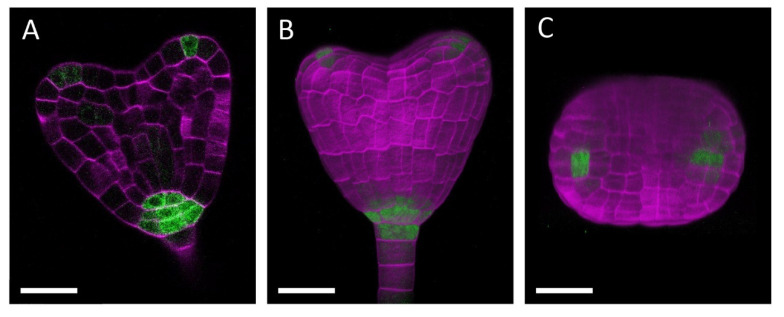
Confocal microscopic images of a ClearSee-processed embryo with a Green Fluorescent Protein (GFP) reporter: (**A**) frontal optical section of a heart stage embryo carrying *DR5rev::GFP* [18] and (**B**,**C**) 3D reconstruction of 157 serial optical sections obtained from the same embryo as in **A** in frontal (**B**) and top (**C**) views. The signals of Calcofluor White and endoplasmic reticulum (ER)-localized GFP are represented in magenta and green, respectively. Bars = 20 μm.

**Figure 3 plants-10-00190-f003:**
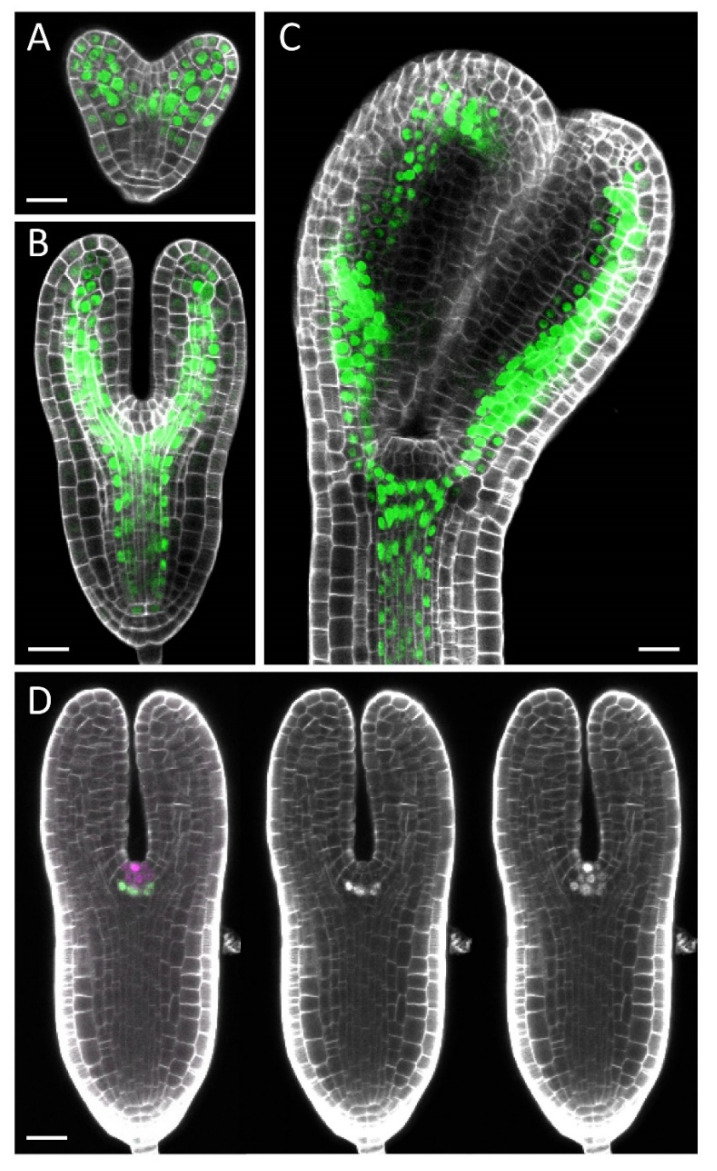
Confocal microscopic images of ClearSee-processed embryos with various fluorescent reporters: (**A**–**C**) frontal optical sections of heart (**A**), torpedo (**B**), and bent-cotyledon (**C**) stage embryos of *pARF5-n3GFP* and (**D**) a frontal optical section of the *gWUS-GFP3 pCLV3:H2B-mScarlet* embryo, displaying signals of the two fluorescent proteins (left), GFP alone (middle), and mScarlet alone (right) together with Calcofluor White signals. In (**A**–**C**) and the left panel of **D**, the signals of Calcofluor White, GFP, and mScarlet are represented in grey scale, green, and magenta, respectively. In the middle and right panels of **D**, the signals of Calcofluor white, GFP, and mScarlet are all represented in grey scale. Bars = 20 μm.

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
