# Peer review of "A ClearSee-Based Clearing Protocol for 3D Visualization of Arabidopsis thaliana Embryos"

_plants, 2021, doi:10.3390/plants10020190_

Round 1

Reviewer 1 Report

The Technical Note by Ayame Imoto and Mitsuhiro Aida, describes ClearSee based clearing protocol, which is suitable for obtaining 3D images of Arabidopsis thaliana embryos. This method enables high quality 3D imaging of embryo morphology and expression of a fluorescent reporter with the cellular resolution.

The paper is written clearly and is easy to read. This protocol described in the paper is designed on a high scientific level.

In my opinion, apart from minor spellcheck and errors in References, this article is ready for the publication as it is.

Please see my comments in the attached file.

Author Response

Thank you very much for your encouraging comments and for precisely checking errors. Those were really helpful.

The Technical Note by Ayame Imoto and Mitsuhiro Aida, describes ClearSee based clearing protocol, which is suitable for obtaining 3D images of Arabidopsis thaliana embryos. This method enables high quality 3D imaging of embryo morphology and expression of a fluorescent reporter with the cellular resolution.

The paper is written clearly and is easy to read. This protocol described in the paper is designed on a high scientific level.

In my opinion, apart from minor spellcheck and errors in References, this article is ready for the publication as it is.

We corrected most errors that you pointed out accordingly. Only one point, the abbreviation of the Nature Method journal is “Nat. Methods” (please see  https://pubmed.ncbi.nlm.nih.gov/22743772/, so we would like to leave that part as in the original version (line 330).

Besides the above corrections, we also did minor revisions: adding a sentence of conclusion to Abstract, correcting several typos. We also added five members to the author list for newly added data in response to comments by Reviewer 2.

Reviewer 2 Report

The manuscript by Imoto and Aida presents a technical update about the adaptation of a ClearSee clearing protocol for Arabidopsis embryo imaging. The text is short and to the point. The interest will be very specific to experts in the field.

General comments: 

  • Please refer to seed, not ovule in the text. "Ovule" is a term for before fertilization. When the ovule is fertilized and an embryo is formed, the used term should be "seed".
  • There are no details on what are the limitations on development stages the protocol can be used. What is the oldest embryonic stage the protocol would be able to clear an embryo?
  • The authors can refer to this new publication in eLife: https://elifesciences.org/articles/63262#s4 where ClearSee cleared ovules were used for morphogenesis analysis.
  • Around lines 80-85, concerning the problems on adherence of embryos to the surfaces. Why not use glass material. One could use well-slides or PAP pen on a slide to define a retention surface. Using then a coated slide and exchanging the solutions instead of transferring the embryos may be a solution. For transferring the embryos, one could use low retention tips or glass capillaries or glass Pasteur pipettes.
  • The embryos are fixed for 10 minutes. In protocols like immuno-staining, fixation is usually 1 hour. How did you verify whether 10 minutes of fixation is enough?
  • Was the protocol tested in a range of GFP-expressing embryos, from low expressed to high expressed reporters? Is the protocol also suitable for other FP?
  • In the material, lines 139, and 143: "described previously". This is a technical report. No one wants to go from reference to reference to find the exact protocol presented and never finding it. So please, describe your exact protocols in this manuscript.

Minor comments:

  • Line 56: "immersed in 7% glucose: what is the volume: is the silique floating or still adhering to the surface? Are the seeds released into the solution? This is a point for the dissection of the seed. Fully immersed would mean the seed is floating, and dissection may be tricky. 
  •  Line 64: Similar comments about the volume of the solution? How do you do to have the embryos not floating away when manipulating other seeds?
  • Line 98: What are the volumes of solution used?
  • Line 109: Why calcofluor white and not SR2200 Renaissance?

Author Response

We really appreciate all your productive comments. Below are our point-by-point responses. Besides them, we also did minor revisions, adding a sentence of conclusion to Abstract and correcting several typos. We also added five members, who were involved in obtaining the new data (Figure 3), to the author list.

The manuscript by Imoto and Aida presents a technical update about the adaptation of a ClearSee clearing protocol for Arabidopsis embryo imaging. The text is short and to the point. The interest will be very specific to experts in the field.

General comments: 

  • Please refer to seed, not ovule in the text. "Ovule" is a term for before fertilization. When the ovule is fertilized and an embryo is formed, the used term should be "seed".

Thank you very much for pointing out this critical error. We overlooked the definitions of ovule and seed. We changed from ovule to seed throughout the text (lines 58 to 102).

  • There are no details on what are the limitations on development stages the protocol can be used. What is the oldest embryonic stage the protocol would be able to clear an embryo?

We newly added data of pARF5-n3GFP, showing that the protocol can be used up to the bent-cotyledon stage (Figure 3A-C; lines 175 to 188 for Results and Discussion; line 209 for Materials and Equipment).

  • The authors can refer to this new publication in eLife: https://elifesciences.org/articles/63262#s4 where ClearSee cleared ovules were used for morphogenesis analysis.

We additionally cited this beautiful paper in Introduction as an example for application of the ClearSee method to plant internal structure (lines 49-51).

  • Around lines 80-85, concerning the problems on adherence of embryos to the surfaces. Why not use glass material. One could use well-slides or PAP pen on a slide to define a retention surface. Using then a coated slide and exchanging the solutions instead of transferring the embryos may be a solution. For transferring the embryos, one could use low retention tips or glass capillaries or glass Pasteur pipettes.

Thank you very much for the suggestions. We did not tested glassware because we overcame the adherence problem by simply adding a small amount of ClearSee to the fixative and washing solutions. We adopted the embryo-transfer strategy instead of solution exchange, because we wanted to avoid accidental drying of the samples, which would greatly hamper the morphology.

  • The embryos are fixed for 10 minutes. In protocols like immuno-staining, fixation is usually 1 hour. How did you verify whether 10 minutes of fixation is enough?

We tested 5-, 10-, 30- and 60-min fixation time, and obtained essentially the same results. We described this in the new version (lines 122-124).

  • Was the protocol tested in a range of GFP-expressing embryos, from low expressed to high expressed reporters? Is the protocol also suitable for other FP?

We have not tested the sensitivity of the methods. Instead, the newly added data using pARF5-n3GFP shows that nuclear localized version of GFP can also be visualized in addition to the ER-localized version used in DR5rev::GFP. Furthermore, we added data of pCLV3::H2B-mScarlet to show that the method could also detect the RFP derivative (Figure 3; lines 175-196 for Results and Discussion; lines 209-228 for Materials and Equipment; line 23 for Abstract; line 54 for Introduction).

  • In the material, lines 139, and 143: "described previously". This is a technical report. No one wants to go from reference to reference to find the exact protocol presented and never finding it. So please, describe your exact protocols in this manuscript.

We added our exact protocols to the corresponding parts (lines 229 to 235; lines 239 to 243).

Minor comments:

  • Line 56: "immersed in 7% glucose: what is the volume: is the silique floating or still adhering to the surface? Are the seeds released into the solution? This is a point for the dissection of the seed. Fully immersed would mean the seed is floating, and dissection may be tricky.

We put 3 ml of 7% glucose in a 35 mm dish (line 67). In our hands, it is very important to let the fruit completely submerged in the solution by holding its pedicel with forceps. In this situation, seeds are also submerged and DO NOT float even after they are excised from the fruit. We now added text to explain the importance of keeping the fruit completely submerged during dissection (lines 67 to 71).

  •  Line 64: Similar comments about the volume of the solution? How do you do to have the embryos not floating away when manipulating other seeds?

At this step, we simply move embryos away from debris in the same 35 mm dish, so the volume of the solution is 3 ml. Embryos DO NOT float after isolation.

  • Line 98: What are the volumes of solution used?

We used 3 ml of ClearSee. We now clarified this in the text (line 130).

  • Line 109: Why calcofluor white and not SR2200 Renaissance?

We used Calcofluor simply because the original ClearSee protocol by Kurihara et al used it. We have not tested SR2200.